# Development and validation of a nomogram to better predict hypertension based on a 10-year retrospective cohort study in China

Xinna Deng[1†], Huiqing Hou[2†], Xiaoxi Wang[2], Qingxia Li[1], Xiuyuan Li[3], Zhaohua Yang[4], Haijiang Wu[4,5]*

[1]Departments of Oncology & Immunotherapy, Hebei General Hospital, Shijiazhuang, China; [2]Physical Examination Center, Hebei General Hospital, Shijiazhuang, China; [3]Department of Foreign Language Teaching, Hebei Medical University, Shijiazhuang, China; [4]Department of Pathology, Hebei Medical University, Shijiazhuang, China; [5]Medical Practice-Education Coordination & Medical Education Research Center, Hebei Medical University, Shijiazhuang, China

*For correspondence:
haijianglaoqi@163.com

[†]These authors contributed equally to this work

Competing interest: The authors declare that no competing interests exist.

## Abstract:

**Background:** Hypertension is a highly prevalent disorder. A nomogram to estimate the risk of hypertension in Chinese individuals is not available.

**Methods:** 6201 subjects were enrolled in the study and randomly divided into training set and validation set at a ratio of 2:1. The LASSO regression technique was used to select the optimal predictive features, and multivariate logistic regression to construct the nomograms. The performance of the nomograms was assessed and validated by AUC, C-index, calibration curves, DCA, clinical impact curves, NRI, and IDI.

**Results:** The nomogram[140/90] was developed with the parameters of family history of hypertension, age, SBP, DBP, BMI, MCHC, MPV, TBIL, and TG. AUCs of nomogram[140/90] were 0.750 in the training set and 0.772 in the validation set. C-index of nomogram[140/90] were 0.750 in the training set and 0.772 in the validation set. The nomogram[130/80] was developed with the parameters of family history of hypertension, age, SBP, DBP, RDWSD, and TBIL. AUCs of nomogram[130/80] were 0.705 in the training set and 0.697 in the validation set. C-index of nomogram[130/80] were 0.705 in the training set and 0.697 in the validation set. Both nomograms demonstrated favorable clinical consistency. NRI and IDI showed that the nomogram[140/90] exhibited superior performance than the nomogram[130/80]. Therefore, the web-based calculator of nomogram[140/90] was built online.

**Conclusions:** We have constructed a nomogram that can be effectively used in the preliminary and in-depth risk prediction of hypertension in a Chinese population based on a 10-year retrospective cohort study.

**Funding:** This study was supported by the Hebei Science and Technology Department Program (no. H2018206110).

# Introduction

Systemic arterial hypertension (hereafter referred to as hypertension) is the most common risk factor for cardiovascular diseases and the biggest contributor to world mortality from noncommunicable diseases (*Mills et al., 2020*; *Burnier and Egan, 2019*). Globally, the number of adults with hypertension increased from 594 million in 1975 to 1.13 billion in 2015; the increase is especially significant

in low-income and middle-income countries (*NCD Risk Factor Collaboration (NCD-RisC), 2017*). As estimated, the number of adults with hypertension is predicted to rise to 1.56 billion by 2025 (*Kearney et al., 2005*). In China, high systolicblood pressure (SBP) is the leading risk factor for both number of deaths and percentage of disability-adjusted life-years, which accounted for 2.54 million deaths in 2017 (*Zhou et al., 2019*). In addition, according to the latest nationwide survey of 451,755 participants from 31 provinces in China, 23.2% (nearly 244.5 million) of Chinese adults have hypertension. The data generated in the China Patient-Centered Evaluative Assessment of Cardiac Events Million Persons Project have shown more serious results [*Joint Committee for Guideline Revision, 2019*]. Among individuals with hypertension, while 46.9% are aware of their condition and 40.7% take prescribed antihypertensive medications, only 15.3% are in control of their blood pressure (*Wang et al., 2018*). Hypertension has imposed so heavy an economic burden on healthcare systems that it requires urgent attention. Early detection of hypertension is vitally important in its control and effective treatment, especially with high-risk subjects.

Hypertension, also known as high blood pressure, is characterized by a persistent elevation of blood pressure in the systemic arteries. Traditionally, the diagnostic criteria of hypertension were SBP ≥140 mmHg and/or diastolicblood pressure (DBP) ≥90 mmHg for the untreated participants, or those taking medication for hypertension. The criteria were broadly accepted by both the 2018 Chinese Guidelines for Prevention and Treatment of Hypertension and the 2018 European Society of Cardiology/European Society of Hypertension guidelines (2018 ESC/ESH) (*Joint Committee for Guideline Revision, 2019* ; *Williams et al., 2018*). However, in November 2017, the American College of Cardiology and the American Heart Association published a guideline for the Prevention, Detection, Evaluation, and Management of High Blood Pressure in Adults (2017 ACC/AHA) (*Whelton et al., 2018*), which redefined the diagnostic criteria of hypertension from 140/90 mmHg to 130/80 mmHg for SBP/DBP. This conspicuous numerically based change results in an increased number of patients being diagnosed with hypertension and in questioning the goal's clinical applicability given the financial burden and clinical outcomes (*López-Jaramillo et al., 2020*). The applicability and potential impact of ACC/AHA 2017 need to be assessed prior to adopting the guideline, especially in China.

Hypertension has been deemed as a complex and multifactorial trait. It is well known that the pathophysiology of hypertension is shaped by combined action of environmental, genetic, anatomical, neural, endocrinal, humoral, and hemodynamic factors (*Rodriguez-Iturbe et al., 2017*). For example, the Dietary Approaches to Stop Hypertension diet is reported to be closely related to lower risk of hypertension (*Navarro-Prado et al., 2020*; *Francisco et al., 2020*). Moreover, psychosocial factors are also possible potentiators and triggers of hypertension. It was showed that psychosocial stress, including occupational stress, socioeconomic pressure, anxiety, and depression, was all associated with greater risk of hypertension, and hypertensive patients had higher level of psychosocial stress compared to normotension patients (*Liu et al., 2017*). Therefore, a simple and reliable model that helps clinicians or subjects to estimate the risk of hypertension is urgently in need.

In the present study, we aimed to develop and validate a risk prediction model for the screening of hypertension by analyzing the routine parameters of physical examination in China.

## Results

### Characteristics of subjects

In Group[140/90], as well as the cut-off value of 140/90 mmHg, the total prevalence of hypertension in 2019 was 24.77% (1536 subjects). At a ratio of 2:1, 4134 subjects were assigned into the training set and 2067 in the validation set. The prevalence of hypertension was 25.35% (1048 subjects) in the training set and 23.61% (488 subjects) in the validation set, respectively. The characteristics of subjects are shown in *Table 1*. There were no significant differences in the characteristics of hypertension status in 2019, gender, family history of hypertension, smoking status, drinking status, age, SBP, DBP, height, weight, body mass index (BMI), white blood cell count (WBC), lymphocyte count (LYMPH), neutrophil count (NEUT), lymphocyte percentage (LYMPHP), neutrophil percentage (NEUTP), red blood cell count (RBC), hemoglobin (HGB), hematocrit (HCT), mean corpuscular volume (MCV), mean corpuscular hemoglobin (MCH), mean cell hemoglobin concentration (MCHC), red blood cell distribution width-coefficient of variation (RDWCV), red blood cell distribution width standard deviation (RDWSD), platelet count (PLT), mean platelet volume (MPV), plateletcrit (PCT), platelet distribution

**Table 1.** Baseline characteristicsh of individuals in training set and validation set of Group[140/90].

| Variables | Training set (N = 4,134) | Validation set (N = 2,067) | P values |
|---|---|---|---|
| Hypertension status in 2019, n (%) | | | |
| No | 3,086 (74.65%) | 1,579 (76.39%) | 0.1342 |
| Yes | 1,048 (25.35%) | 488 (23.61%) | |
| Gender | | | |
| Female | 1,899 (45.94%) | 968 (46.83%) | 0.5052 |
| Male | 2,235 (54.06%) | 1,099 (53.17%) | |
| Family history of hypertension | | | |
| No | 3,029 (73.27%) | 1,491 (72.13%) | 0.3424 |
| Yes | 1,105 (26.73%) | 576 (27.87%) | |
| Smoking status | | | |
| No | 3,846 (93.03%) | 1945 (94.10%) | 0.1118 |
| Yes | 288 (6.97%) | 122 (5.90%) | |
| Drinking status | | | |
| No | 3,715 (89.86%) | 1,871 (90.52%) | 0.4173 |
| Yes | 419 (10.14%) | 196 (9.48%) | |
| Age, year | 45.00 (37.00,54.00) | 45.00 (37.00,54.00) | 0.5448 |
| SBP,mmHg | 110.00 (100.00,120.00) | 110.00 (100.00,120.00) | 0.1163 |
| DBP,mmHg | 70.00 (70.00,80.00) | 70.00 (70.00,80.00) | 0.2938 |
| Height, cm | 167.00 (161.00,173.00) | 166.00 (161.00,173.00) | 0.3414 |
| weight, kg | 66.00 (59.00,76.00) | 66.00 (59.00,75.00) | 0.8163 |
| BMI, kg/m$^2$ | 24.03 (21.89,26.22) | 24.07 (21.89,26.09) | 0.9343 |
| WBC, 10$^9$/L | 5.50 (4.60,6.40) | 5.50 (4.60,6.30) | 0.6647 |
| LYMPH, 10$^9$/L | 1.80 (1.50,2.10) | 1.80 (1.50,2.10) | 0.9866 |
| NEUT, 10$^9$/L | 3.20 (2.60,3.90) | 3.20 (2.60,3.80) | 0.7023 |
| LYMPHP, % | 33.20 (28.90,37.80) | 33.60 (29.20,37.90) | 0.2718 |
| NEUTP, % | 58.50 (53.70,63.20) | 58.30 (53.70,63.10) | 0.6369 |
| RBC, 10$^{12}$/L | 4.25 (3.93,4.57) | 4.25 (3.93,4.56) | 0.9433 |
| HGB, g/L | 130.00 (119.00,141.00) | 129.00 (119.00,141.00) | 0.6263 |
| HCT, % | 39.20 (37.00,42.30) | 39.10 (37.00,42.10) | 0.5409 |
| MCV, fL | 92.20 (89.70,94.60) | 92.20 (89.60,94.60) | 0.5249 |
| MCH, pg | 30.60 (29.60,31.00) | 30.60 (29.60,31.00) | 0.5454 |
| MCHC, g/L | 331.00 (325.00,337.00) | 331.00 (325.00,337.00) | 0.5947 |
| RDWCV, % | 14.40 (14.00,14.50) | 14.40 (14.00,14.50) | 0.1879 |
| RDWSD, fL | 48.70 (46.50,50.20) | 48.70 (46.50,50.20) | 0.8534 |
| PLT, 10$^9$/L | 211.00 (182.00,243.00) | 211.00 (184.00,243.00) | 0.9817 |
| MPV, fL | 8.80 (8.30,9.30) | 8.80 (8.30,9.30) | 0.3084 |
| PCT, % | 0.18 (0.16,0.21) | 0.19 (0.16,0.21) | 0.5154 |
| PDW, fL | 15.80 (15.70,16.00) | 15.80 (15.70,16.00) | 0.8825 |
| MID, 10$^9$/L | 0.40 (0.30,0.50) | 0.40 (0.30,0.50) | 0.1560 |
| MIDP, % | 8.20 (7.10,9.00) | 8.10 (7.10,9.00) | 0.1760 |
| ALT, U/L | 17.90 (14.20,24.30) | 18.00 (14.30,23.90) | 0.8201 |
| AST, U/L | 19.40 (17.00,22.90) | 19.40 (17.00,22.60) | 0.6713 |

*Table 1 continued on next page*

*Table 1 continued*

| Variables | Training set (N = 4,134) | Validation set (N = 2,067) | P values |
|---|---|---|---|
| TP, g/L | 71.40 (69.00,73.90) | 71.40 (69.00,73.90) | 0.8951 |
| ALB, g/L | 42.38 (40.82,44.18) | 42.44 (40.83,44.00) | 0.4247 |
| TBIL, µmol/L | 16.90 (13.80,20.80) | 16.80 (14.00,20.80) | 0.6959 |
| DBIL, µmol/L | 2.00 (2.00,3.00) | 2.00 (2.00,3.00) | 0.9172 |
| GLU, mmol/L | 5.49 (5.19,5.86) | 5.50 (5.17,5.87) | 0.7616 |
| CHOL, mmol/L | 4.71 (4.17,5.32) | 4.72 (4.19,5.34) | 0.3311 |
| TG, mmol/L | 1.20 (0.82,1.78) | 1.20 (0.82,1.78) | 0.9119 |
| NLR, % | 1.76 (1.43,2.19) | 1.74 (1.42,2.17) | 0.2978 |
| PLR, % | 116.75 (96.07,141.88) | 117.06 (97.50,140.67) | 0.7375 |

Data are presented as median (25% percentile, 75% percentile) for continuous variables and count (percentage) for categorical variables.

SBP: systolic blood pressure; DBP: diastolic blood pressure; BMI: body mass index; WBC: white blood cell count; LYMPH: lymphocyte count; NEUT: neutrophil count; LYMPHP: lymphocyte percentage; NEUTP: neutrophil percentage; RBC: red blood cell count; HGB: hemoglobin; HCT: hematocrit; MCV: mean corpuscular volume; MCH: mean corpuscular hemoglobin; MCHC: mean cell hemoglobin concentration; RDWCV: red blood cell distribution width-coefficient of variation; RDWSD: red blood cell distribution width standard deviation; PLT: platelet count; MPV: mean platelet volume; PCT: plateletcrit; PDW: platelet distribution width; MID: middle cell count; MIDP: middle cell percentage; ALT: alanine aminotransferase; AST: aspartate transaminase; TP: total protein; ALB: albumin; TBIL: total bilirubin; DBIL: direct bilirubin; GLU: glucose; CHOL: cholesterol; TG: triglycerides; NLR: neutrophil-to-lymphocyte ratio; PLR: platelet-to-lymphocyte ratio.

width (PDW), middle cell count (MID), middle cell percentage (MIDP), alanine aminotransferase (ALT), aspartate transaminase (AST), total protein (TP), albumin (ALB), total bilirubin (TBIL), direct bilirubin (DBIL), glucose (GLU), cholesterol (CHOL), triglycerides (TG), neutrophil-to-lymphocyte ratio (NLR), and platelet-to-lymphocyte ratio (PLR) between the two sets.

In Group[130/80], as well as the cut-off value of 130/80 mmHg, the total prevalence of hypertension in 2019 was 37.92% (1430 subjects). At a ratio of 2:1, 2514 subjects were assigned into the training set and 1257 in the validation set. The prevalence of hypertension was 37.39% (940 subjects) in the training set and 38.98% (490 subjects) in the validation set, respectively. The characteristics of subjects are shown in *Table 2*. There were no significant differences in characteristics of hypertension status in 2019, gender, family history of hypertension, smoking status, drinking status, age, SBP, DBP, height, weight, BMI, WBC, LYMPH, NEUT, LYMPHP, NEUTP, RBC, HGB, HCT, MCV, MCH, MCHC, RDWCV, RDWSD, PLT, MPV, PCT, PDW, MID, MIDP, ALT, AST, TP, ALB, TBIL, DBIL, GLU, CHOL, TG, NLR, and PLR between two sets.

## Construction of nomogram[140/90] and nomogram[130/80]

In Group[140/90], 21 variables had nonzero coefficients in the least absoluteshrinkage and selection operator (LASSO) regression model based on the analysis of the training set. These variables included gender, family history of hypertension, age, SBP, DBP, height, BMI, LYMPH, LYMPHP, NEUTP, MCHC, RDWCV, RDWSD, PLT, MPV, AST, TP, TBIL, GLU, TG, and NLR (*Figure 1A B*). Multivariate logistic regression analysis revealed that family history of hypertension, age, SBP, DBP, BMI, MCHC, MPV, TP, TBIL, and TG were independent risk factors for hypertension (*Table 3*). These 10 independent factors were used to construct the nomogram[140/90] (*Figure 2A*).

In Group[130/80], 21 variables had nonzero coefficients in the LASSO regression model based on the analysis of the training set. These variables included family history of hypertension, drinking status, age, SBP, DBP, weight, BMI, WBC, NEUT, LYMPHP, RBC, RDWSD, PCT, PDW, ALT, AST, TP, TBIL, GLU, CHOL, and TG (*Figure 1C D*). Multivariate logistic regression analysis revealed that family history of hypertension, age, SBP, DBP, RDWSD, and TBIL were independent risk factors for hypertension (*Table 4*). These six independent factors were used to construct the nomogram[130/80] (*Figure 2B*).

## Assessment of nomogram[140/90] and nomogram[130/80] in the training set

As mentioned above, the nomogram[140/90] was constructed to predict the risk of hypertension by using family history of hypertension, age, SBP, DBP, BMI, MCHC, MPV, TP, TBIL, and TG. Its performance was assessed with the area under receiver operating characteristic curve (AUC) and C-index. The AUC value of the nomogram[140/90] was 0.750 (95% CI: 0.733–0.767). The AUCs of family history of hypertension, age, SBP, DBP, BMI, MCHC, MPV, TP, TBIL, and TG were 0.554 (95% CI: 0.534–0.575),

**Table 2.** Baseline characteristics of individuals in training set and validation set of Group[130/80].

| Variables | Training set (N = 2,514) | Validation set (N = 1,257) | P values |
|---|---|---|---|
| **Hypertension status in 2019, n (%)** | | | |
| No | 1,574 (62.61%) | 767 (61.02%) | 0.3425 |
| Yes | 940 (37.39%) | 490 (38.98%) | |
| **Gender** | | | |
| Female | 1,436 (57.12%) | 685 (54.49%) | 0.1255 |
| Male | 1,078 (42.88%) | 572 (45.51%) | |
| **Family history of hypertension** | | | |
| No | 1,886 (75.02%) | 949 (75.50%) | 0.7491 |
| Yes | 628 (24.98%) | 308 (24.50%) | |
| **Smoking status** | | | |
| No | 2,403 (95.58%) | 1,195 (95.07%) | 0.4743 |
| Yes | 111 (4.42%) | 62 (4.93%) | |
| **Drinking status** | | | |
| No | 2,342 (93.16%) | 1,173 (93.32%) | 0.8547 |
| Yes | 172 (6.84%) | 84 (6.68%) | |
| Age, year | 43.00 (36.00,53.00) | 44.00 (37.00,53.00) | 0.2488 |
| SBP, mmHg | 104.00 (100.00,110.00) | 104.00 (100.00,110.00) | 0.3309 |
| DBP, mmHg | 70.00 (62.00,70.00) | 70.00 (64.00,70.00) | 0.1694 |
| Height, cm | 165.00 (160.00,172.00) | 165.00 (160.00,172.00) | 0.8938 |
| weight, kg | 63.00 (56.00,72.00) | 63.00 (56.00,72.00) | 0.3730 |
| BMI, kg/m$^2$ | 23.24 (21.30,25.39) | 23.24 (21.09,25.27) | 0.3171 |
| WBC, 10$^9$/L | 5.30 (4.50,6.30) | 5.30 (4.50,6.30) | 0.7810 |
| LYMPH, 10$^9$/L | 1.80 (1.50,2.10) | 1.70 (1.50,2.10) | 0.3251 |
| NEUT, 10$^9$/L | 3.10 (2.50,3.80) | 3.10 (2.50,3.80) | 0.9247 |
| LYMPHP, % | 33.40 (29.00,37.90) | 33.40 (29.20,38.20) | 0.5755 |
| NEUTP, % | 58.30 (53.70,63.20) | 58.50 (53.25,63.20) | 0.6634 |
| RBC, 10$^{12}$/L | 4.15 (3.87,4.48) | 4.14 (3.86,4.47) | 0.4514 |
| HGB, g/L | 126.00 (116.00,138.00) | 126.00 (117.00,138.00) | 0.5630 |
| HCT, % | 38.10 (37.00,41.30) | 38.20 (37.00,41.40) | 0.6286 |
| MCV, fL | 92.20 (89.50,94.60) | 92.30 (89.75,94.80) | 0.2552 |
| MCH, pg | 30.50 (29.40,31.00) | 30.60 (29.55,31.00) | 0.0824 |
| MCHC, g/L | 330.00 (324.00,336.00) | 331.00 (324.00,337.00) | 0.0561 |
| RDWCV, % | 14.50 (14.00,14.50) | 14.50 (14.10,14.50) | 0.9732 |
| RDWSD, fL | 48.70 (47.20,50.20) | 48.70 (47.20,50.20) | 0.1531 |
| PLT, 10$^9$/L | 210.00 (181.00,241.00) | 210.00 (181.00,239.50) | 0.8556 |
| MPV, fL | 8.80 (8.30,9.30) | 8.90 (8.40,9.30) | 0.3158 |
| PCT, % | 0.18 (0.16,0.21) | 0.18 (0.16,0.21) | 0.9297 |
| PDW, fL | 15.80 (15.68,16.00) | 15.80 (15.70,16.00) | 0.6276 |
| MID, 10$^9$/L | 0.40 (0.30,0.50) | 0.40 (0.30,0.50) | 0.5171 |

*Table 2 continued on next page*

Table 2 continued

| Variables | Training set (N = 2,514) | Validation set (N = 1,257) | P values |
|---|---|---|---|
| MIDP, % | 8.10 (7.10,9.00) | 8.10 (6.90,9.00) | 0.5763 |
| ALT, U/L | 16.60 (13.50,22.20) | 16.70 (13.50,22.30) | 0.5375 |
| AST, U/L | 18.70 (16.50,22.20) | 18.90 (16.50,22.00) | 0.5839 |
| TP, g/L | 71.30 (69.00,73.90) | 71.20 (68.80,73.60) | 0.3776 |
| ALB, g/L | 42.21 (40.64,43.96) | 42.29 (40.64,43.96) | 0.4390 |
| TBIL, µmol/L | 16.70 (13.70,20.50) | 16.70 (13.70,20.45) | 0.6017 |
| DBIL, µmol/L | 2.00 (2.00,3.00) | 2.00 (2.00,3.00) | 0.8340 |
| GLU, mmol/L | 5.43 (5.15,5.77) | 5.42 (5.13,5.73) | 0.3640 |
| CHOL, mmol/L | 4.65 (4.10,5.24) | 4.69 (4.13,5.32) | 0.1182 |
| TG, mmol/L | 1.05 (0.74,1.56) | 1.06 (0.75,1.60) | 0.4149 |
| NLR, % | 1.75 (1.43,2.19) | 1.75 (1.41,2.17) | 0.5353 |
| PLR, % | 119.23 (98.57,144.44) | 117.62 (96.10,143.43) | 0.1833 |

Data were presented as median (the 25% percentile, the 75% percentile) for continuous variables and count (percentage) for categorical variables.
SBP, systolic blood pressure; DBP, diastolic blood pressure; BMI, body mass index; WBC, white blood cell count; LYMPH, lymphocyte count; NEUT, neutrophil count; LYMPHP, lymphocyte percentage; NEUTP, neutrophil percentage; RBC, red blood cell count; HGB, hemoglobin; HCT, hematocrit; MCV, mean corpuscular volume; MCH, mean corpuscular hemoglobin; MCHC, mean cell hemoglobin concentration; RDWCV, red blood cell distribution width-coefficient of variation; RDWSD, red blood cell distribution width standard deviation; PLT, platelet count; MPV, mean platelet volume; PCT, plateletcrit; PDW, platelet distribution width; MID, middle cell count; MIDP, middle cell percentage; ALT, alanine aminotransferase; AST, aspartate transaminase; TP, total protein; ALB, albumin; TBIL, total bilirubin; DBIL, direct bilirubin; GLU, glucose; CHOL, cholesterol; TG, triglycerides; NLR, neutrophil to lymphocyte ratio; PLR, platelet-to-lymphocyte ratio.

0.659 (95% CI: 0.640–0.677), 0.711 (95% CI: 0.693–0.729), 0.645 (95% CI: 0.626–0.664), 0.618 (95% CI: 0.598–0.637), 0.513 (95% CI: 0.492–0.533), 0.507 (95% CI: 0.486–0.527), 0.515 (95% CI: 0.495–0.535), 0.517 (95% CI: 0.496–0.537), and 0.605 (95% CI: 0.586–0.625), respectively (*Figure 3A*). The C-index of nomogram[140/90] was 0.75 (95% CI: 0.733–0.767) in the training set, which indicated that the model had a good predictive discrimination. Furthermore, the calibration curve showed a high consistency between prediction and actual observation (*Figure 4A*).

As mentioned above, the nomogram[130/80] was constructed to predict the risk of hypertension with family history of hypertension, age, SBP, DBP, RDWSD, and TBIL. The AUC value of the nomogram[130/80] was 0.705 (95% CI: 0.684–0.725). The AUCs of family history of hypertension, age, SBP, DBP, RDWSD, and TBIL were 0.523 (95% CI: 0.500–0.547), 0.629 (95% CI: 0.607–0.652), 0.669 (95% CI: 0.648–0.691), 0.623 (95% CI: 0.600–0.645), 0.532 (95% CI: 0.509–0.555), and 0.523 (95% CI: 0.500–0.547), respectively (*Figure 3B*). The C-index of nomogram[130/80] was 0.705 (95% CI: 0.684–0.726) in the training set, which indicated that the model had a relatively bad predictive discrimination. Furthermore, the calibration curve showed a relatively low consistency between prediction and actual observation (*Figure 4B*).

## Validation of nomogram[140/90] and nomogram[130/80] in the validation set

Since the data of the training set were used to construct the nomogram, the data of the validation set were employed to validate the nomogram. The AUC of nomogram[140/90] was 0.772 (95% CI: 0.749–0.795) (*Figure 3C*). The C-index of nomogram[140/90] was 0.772 (95% CI: 0.748–0.796). The calibration curve also showed a high consistency between prediction and actual observation (*Figure 4C*). The AUC of nomogram[130/80] was 0.697 (95% CI: 0.668–0.726) (*Figure 3D*). The C-index of nomogram[130/80] was 0.697 (95% CI: 0.668–0.726). The calibration curve also showed a relatively low consistency between prediction and actual observation (*Figure 4D*).

## Clinical utility of nomogram[140/90] and nomogram[130/80]

The decision curve analysis (DCA) showed that the nomogram[140/90] and nomogram[130/80] had greater net benefits for the identification of hypertension than that of any single factor in the training sets, respectively (*Figure 5A B*). Similar results were found in the validation sets (*Figure 5C D*). In addition,

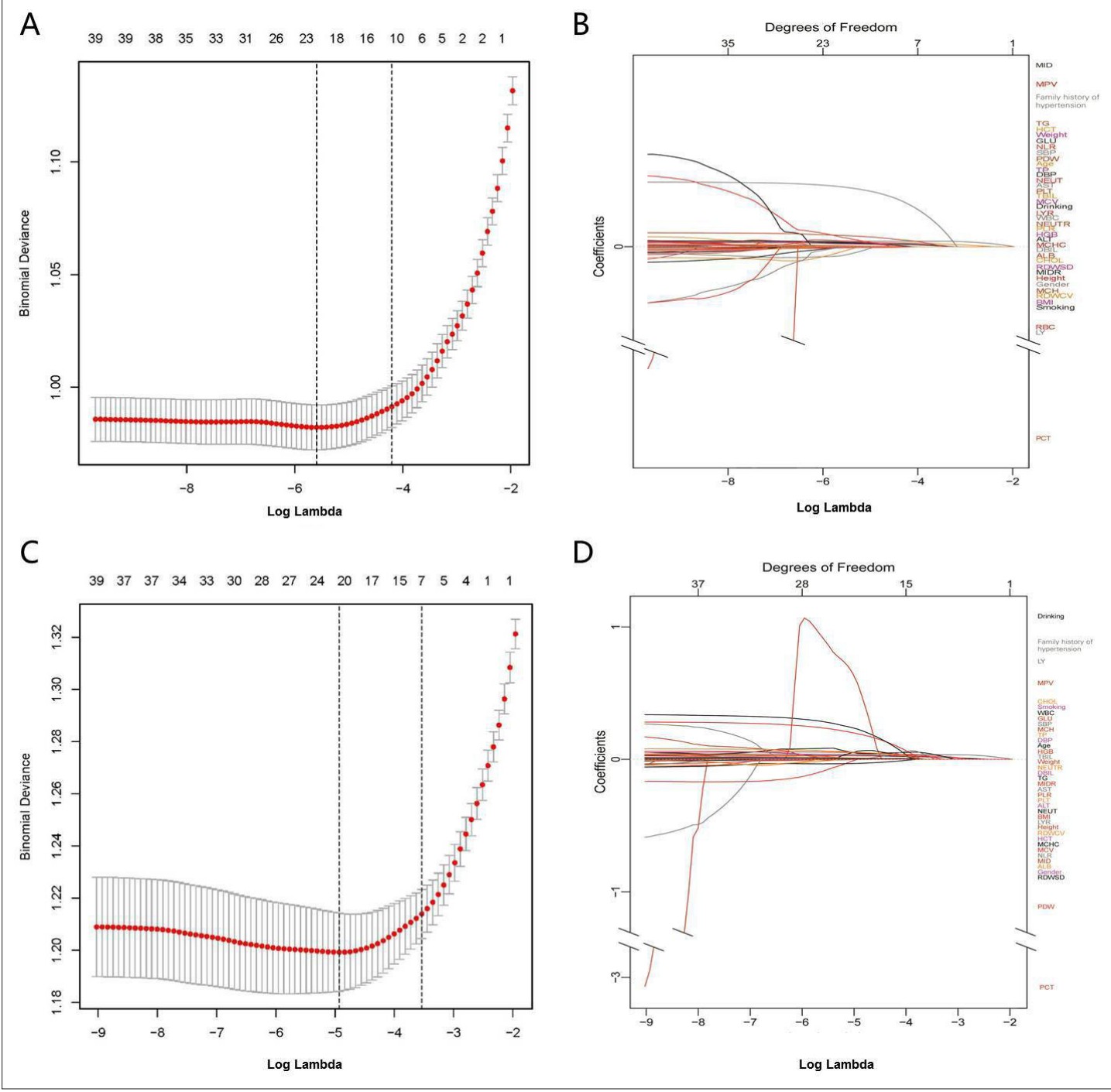

**Figure 1.** Texture feature selection using the least absolute shrinkage and selection operator (LASSO) binary logistic regression model. (**A**) Identification of the optimal penalization coefficient lambda ($\lambda$) in the LASSO model with 10-fold cross-validation in Group[140/90]. (**B**) LASSO coefficient profiles of 21 features in Group[140/90]. The trajectory of each hypertension-related features' coefficient was observed in the LASSO coefficient profiles with the changing of the lambda in LASSO algorithm. (**C**) Identification of the optimal penalization coefficient lambda ($\lambda$) in the LASSO model with 10-fold cross-validation in Group[130/80]. (**D**) LASSO coefficient profiles of 21 features in Group[130/80]. The trajectory of each hypertension-related features' coefficient was observed in the LASSO coefficient profiles with the changing of the lambda in LASSO algorithm. SBP: systolic blood pressure; DBP: diastolic blood pressure; BMI: body mass index; WBC: white blood cell count; LYMPH: lymphocyte count; NEUT: neutrophil count; LYMPHP: lymphocyte percentage; NEUTP: neutrophil percentage; RBC: red blood cell count; MCHC: mean cell hemoglobin concentration; RDWCV: red blood cell distribution width-coefficient of variation; RDWSD: red blood cell distribution width standard deviation; PLT: platelet count; MPV: mean platelet volume; PCT: plateletcrit; PDW: platelet distribution width; ALT: alanine aminotransferase; AST: aspartate transaminase; TP: total protein; TBIL: total bilirubin; GLU: glucose; CHOL: cholesterol; TG: triglycerides; NLR: neutrophil-to-lymphocyte ratio.

**Table 3.** Risk factors for hypertension in the training set of Group[140/90].

| Variable | Model β-Coefficient | Odds ratio (95%CI) | p value |
|---|---|---|---|
| **Family history of hypertension** | | | |
| No | Reference | | |
| Yes | 0.483 | 1.621 (1.372–1.913) | <0.001 |
| Age | 0.036 | 1.037 (1.028–1.045) | <0.001 |
| SBP | 0.041 | 1.041 (1.032–1.051) | <0.001 |
| DBP | 0.031 | 1.031 (1.017–1.046) | <0.001 |
| BMI | 0.039 | 1.040 (1.011–1.069) | 0.006 |
| MCHC | –0.015 | 0.985 (0.975–0.995) | 0.003 |
| MPV | 0.161 | 1.175 (1.036–1.333) | 0.012 |
| TP | 0.025 | 1.025 (1.003–1.049) | 0.028 |
| TBIL | 0.015 | 1.015 (1.002–1.028) | 0.023 |
| TG | 0.098 | 1.102 (0.962–1.132) | 0.012 |

SBP: systolic blood pressure; DBP: diastolic blood pressure; BMI: body mass index; MCHC: mean cell hemoglobinconcentration; MPV: mean platelet volume; TP: total protein; TBIL: total bilirubin; TG: triglycerides.

based on the results of DCA, we further plotted clinical impact curves to evaluate the clinical utility of the nomograms. The clinical impact curves of nomogram[140/90] and nomogram[130/80] showed that the predicted probability coincided well with the actual probability in the training sets, respectively (*Figure 6A B*). Similar results were found in the validation sets (*Figure 6C D*).

## Comparison between nomogram[140/90] and nomogram[130/80]

In Group[130/80], compared with the nomogram[130/80], the nomogram[140/90] resulted in a categorical net reclassification improvement (NRI) of 0.0081 (95% CI: −0.0097–0.0258, p=0.372), continuous NRI of 0.1174 (95% CI: 0.0517–0.1831, p<0.001), and integrated discrimination improvement (IDI) of 0.0032 (95% CI: 0.0001–0.0063, p=0.0432) for predicting the risk of hypertension. These results indicated that the nomogram[140/90] exhibited superior predictive capability than nomogram[130/80].

## Website of nomogram[140/90]

The web-based user-friendly calculator of nomogram[140/90] (https://haijianglaoqi.shinyapps.io/Risk-of-hypertension/) was developed and freely available online to help patients and physicians to calculate the risk of hypertension.

## Discussion

Accurate and timely diagnosis of hypertension is of great importance for effective therapy. Therefore, it is necessary to establish a model to estimate the risk of hypertension to aid in risk stratification and management. Some hypertension risk prediction models have been preliminarily developed in different populations in the past decade, such as in Iranian, Korean, Japanese, and Indian (*Bozorg-manesh et al., 2011*; *Lim et al., 2013*; *Otsuka et al., 2015*; *Sathish et al., 2016*). In these models, the risk factors of hypertension varied widely across studies. Age, SBP, DBP, and current smoking status were the most common independent factors in the studied population, and all of them were included in four prediction models. However, there is no agreement among investigators as to what constitutes a major predictor. It is therefore suggested that a hypertension risk prediction model developed in the particular racial, ethnic, or national groups may not be directly applied to other populations.

In China, some hypertension risk prediction models were also established. In 2016, *Chen et al., 2016* constructed a sex-specific multivariable hypertension prediction model based on northern urban Han Chinese population. The predictive model yielded an AUC of 0.761 for men and 0.753

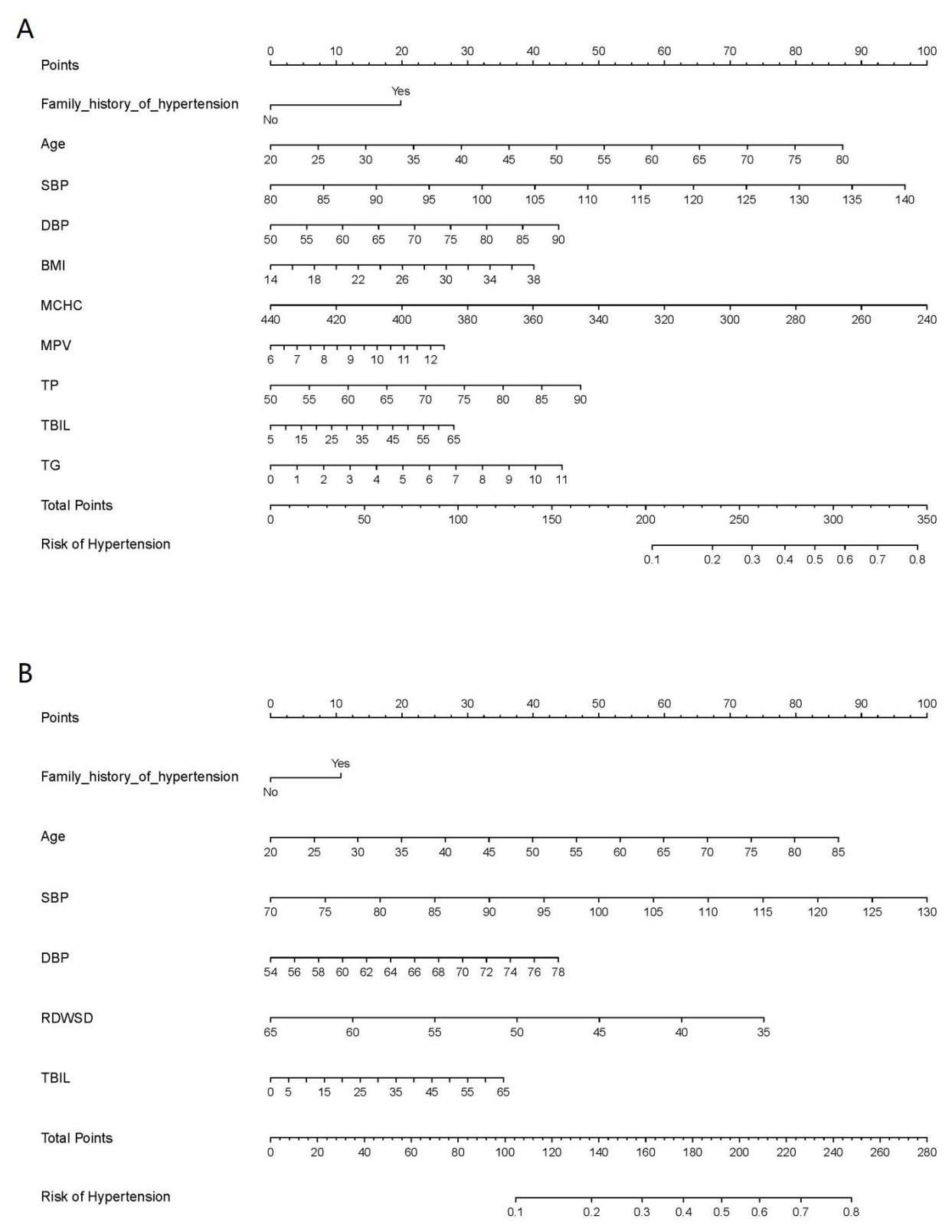

**Figure 2.** Nomogram for the prediction of hypertension. (**A**) Nomogram[140/90] was constructed based on the data of Group[140/90]. (**B**) Nomogram[130/80] was constructed based on the data of Group[130/80]. The points of each features were added to obtain the total points, and a vertical line was drawn on the total points to obtain the corresponding 'risk of hypertension'. SBP: systolic blood pressure; DBP: diastolic blood pressure; BMI: body mass index; MCHC: mean cell hemoglobin concentration; MPV: mean platelet volume; TP: total protein; TBIL: total bilirubin; TG: triglycerides; RDWSD: red blood cell distribution width standard deviation.

**Table 4.** Risk factors for hypertension in the training set of Group[130/80].

| Variable | Model | | |
| --- | --- | --- | --- |
| | β-Coefficient | Odds ratio (95%CI) | p value |
| **Family history of hypertension** | | | |
| No | Reference | | |
| Yes | 0.282 | 1.326 (1.084–1.620) | 0.006 |
| Age | 0.032 | 1.032 (1.022–1.042) | <0.001 |
| SBP | 0.040 | 1.041 (1.029–1.053) | <0.001 |
| DBP | 0.033 | 1.034 (1.012–1.065) | 0.002 |
| RDWSD | −0.047 | 0.954 (0.916–0.994) | 0.024 |
| TBIL | 0.015 | 1.016 (1.001–1.031) | 0.041 |

SBP: systolic blood pressure; DBP: diastolic blood pressure; RDWSD: red blood celldistribution width standard deviation; TBIL: total bilirubin.

for women. The limitation of their study is that it did not perform internal or external validation. In 2019, *Xu et al., 2019* constructed several predictive models for hypertension among Chinese rural populations. In the training set, AUCs ranged from 0.720 to 0.767 for men and from 0.740 to 0.809 for women. In the testing set, AUCs ranged from 0.722 to 0.773 for men and from 0.698 to 0.765 for women. Two studies mentioned above were carried out either in a single rural area or urban area. Another predicted model for hypertension based on a large cross-sectional study was established recently (*Ren et al., 2020*). In spite of a large group of people (73,158 samples), the prediction performances of the model were simply assessed by probability of disease (POD) index and AUC values (76.52% in the train set and 75.81% in the test set). Therefore, the present study might be the first one to develop nomogram for the prediction of hypertension based on systematic assessment and validation in China.

In this study, according to two diagnostic criteria of hypertension, we developed and validated two nomograms for the prediction of hypertension based on a 10-year retrospective cohort study in Chinese population. Both nomograms were constructed mainly based on the physical examination data. The nomogram[140/90] incorporated 10 parameters including family history of hypertension, age, SBP, DBP, BMI, MCHC, MPV, TP, TBIL, and TG. The nomogram[130/80] incorporated six parameters including family history of hypertension, age, SBP, DBP, RDWSD, and TBIL. All parameters are readily available in routine health examinations. Therefore, these nomograms will be useful for the in-depth assessment without the assistance of physicians. Notably, receiver operating characteristic (ROC) analysis indicated that AUC of nomogram[140/90] was higher than that of nomogram[130/80]. The nomogram[140/90] also displayed excellent discrimination with a C-index of 0.75 and good calibration. High C-index value of 0.772 could still be reached in the internal validation. DCA and clinical impact curve showed that the majority of the threshold probabilities in this model had good net benefits. Moreover, NRI and IDI were originally proposed to characterize accuracy improvement in predicting a binary outcome, when new biomarkers are added to regression models. Most recently, these two indices have been extended from binary outcomes to multicategorical and survival outcomes (*Wang et al., 2020*). For example, in 2020, Zhang et al. compared the predictive ability of a stroke prediction model (China-PAR) with the revised Framingham Stroke Risk Score (R-FSRS) for 5-year stroke incidence in a community cohort of Chinese adults [*Zhang et al., 2020*]. The two prediction models have five same risk factors, as well as two additional factors in R-FSRS and six additional factors in China-PAR. The NRI and IDI values were assessed to compare the discrimination ability of two prediction models. Similarly, to better assess the performance of nomogram[140/90] and nomogram[130/80], the NRI and IDI were also used to determine the best model. In this study, the NRI and IDI showed that the nomogram[140/90] exhibited superior performance than nomogram[130/80]. Thus, the nomogram[140/90] is the most sensitive hypertension risk prediction tool under the promise of guaranteeing accuracy.

Multivariate logistic regression analysis revealed that gender was not an independent predictor for hypertension in our study. Similar to our study, researchers from Iran revealed the same result,

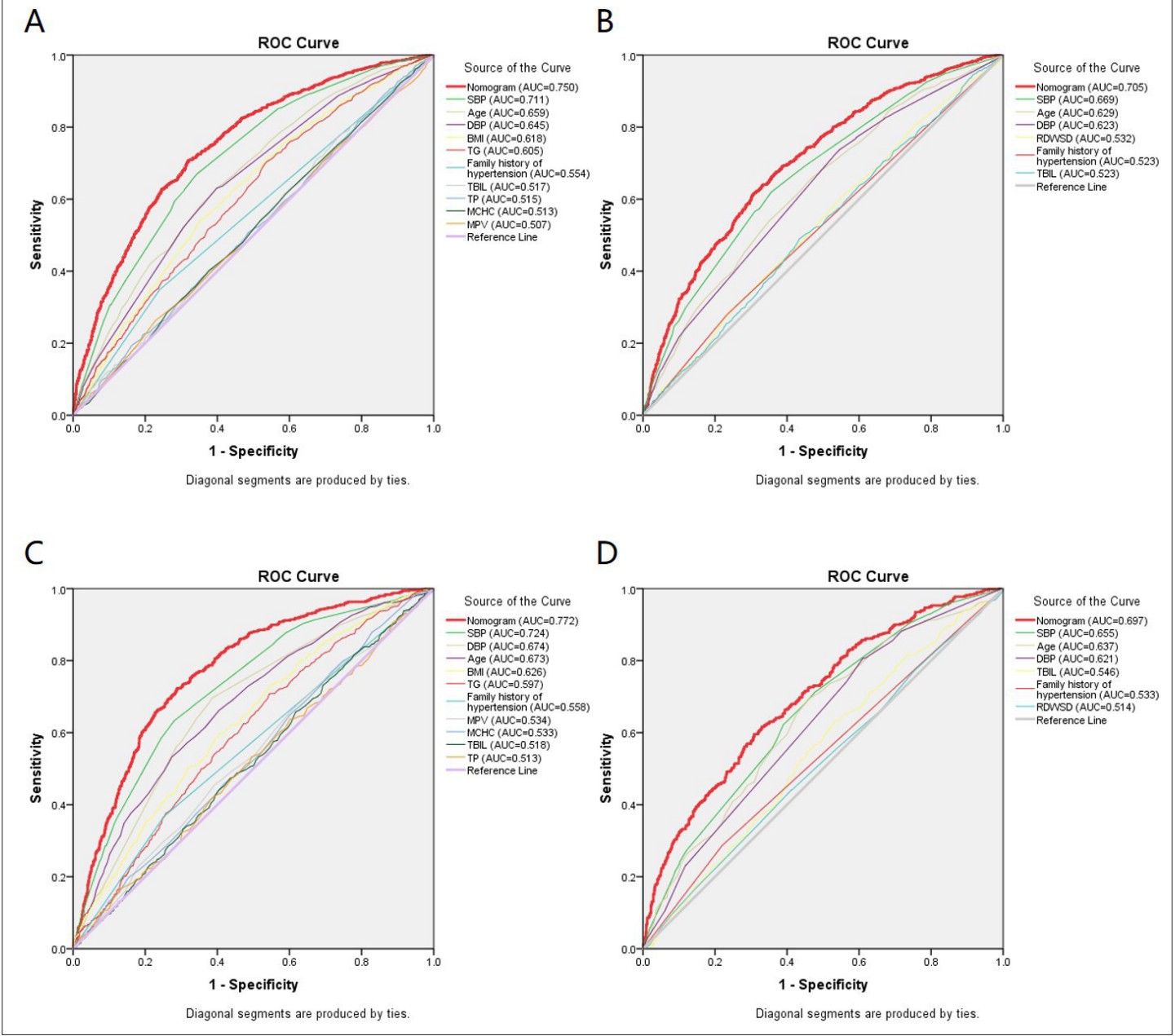

**Figure 3.** Receiver operating characteristic (ROC) curves for the prediction of hypertension in the training set and validation set. (**A**) ROC curves of the factors and nomogram[140/90] in the training set of Group[140/90]. (**B**) ROC curves of the factors and nomogram[130/80] in the training set of Group[130/80]. (**C**) ROC curves of the factors and nomogram[140/90] in the validation set of Group[140/90]. (**D**) ROC curves of the factors and nomogram[130/80] in the validation set of Group[130/80]. SBP: systolic blood pressure; DBP: diastolic blood pressure; BMI: body mass index; MCHC: mean cell hemoglobin concentration; MPV: mean platelet volume; TP: total protein; TBIL: total bilirubin; TG: triglycerides; RDWSD: red blood cell distribution width standard deviation.

reporting that sex was not found to be an independent risk factor for hypertension (*Talaei et al., 2014*). Nonetheless, several previous studies have reported gender to be significantly associated with hypertension (*Chen et al., 2016*; *Fidalgo et al., 2019*; *Zheng et al., 2014*; *Kshirsagar et al., 2010*). It is still controversial whether the incidence of hypertension is associated with gender. In the future, the role of gender on blood pressure and its correlation with hypertension need further evaluation with larger population cohorts.

In our study, the total incidence of hypertension was 24.77% and 37.92% at a cut-off value of 140/90 mmHg and 130/80 mmHg, respectively. The reported prevalence of hypertension in China shows a great geographical variation, ranging from 23.2% to 44.7% (*Joint Committee for Guideline*

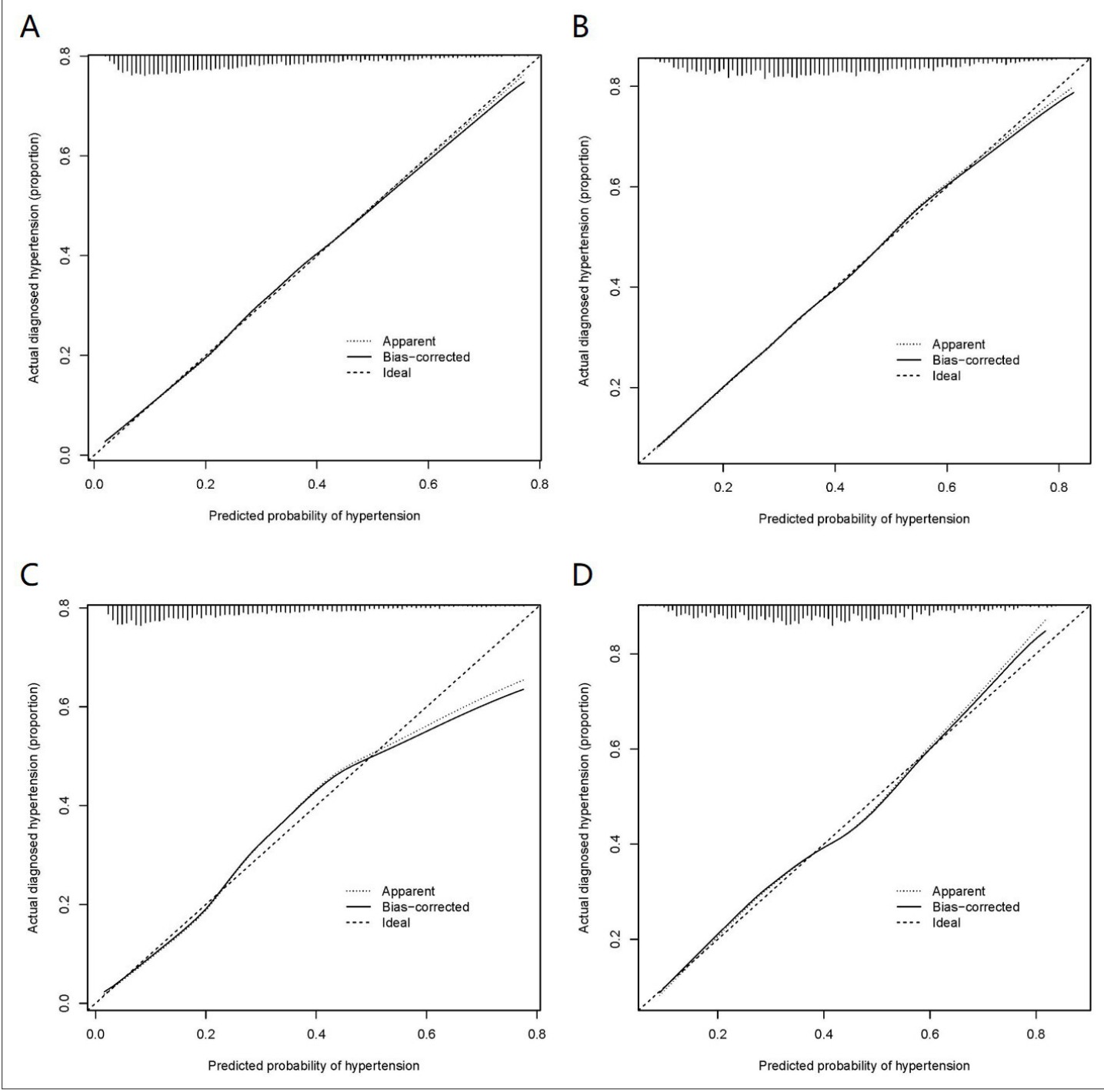

**Figure 4.** Calibration curves of the nomogram prediction in the training set and validation set. (**A**) Calibration curves of nomogram$^{140/90}$ prediction in the training set of Group$^{140/90}$. (**B**) Calibration curves of nomogram$^{130/80}$ prediction in the training set of Group$^{130/80}$. (**C**) Calibration curves of nomogram$^{140/90}$ prediction in the validation set of Group$^{140/90}$. (**D**) Calibration curves of nomogram$^{130/80}$ prediction in the validation set of Group$^{130/80}$.

*Revision, 2019*; *Lu et al., 2017*; *Asgari et al., 2020*; *Labasangzhu et al., 2020*). The difference between these studies may be caused by the following reasons. Firstly, the surveys mentioned above were conducted in different periods and in different age groups by different organizations, which may result in inconsistencies. Secondly, different dietary habits and lifestyles among different populations may contribute to observed differences. For example, mean sodium intake of northern Chinese was notably higher than that of southerners [*Heizhati et al., 2020*]. That excessive salt intake increases

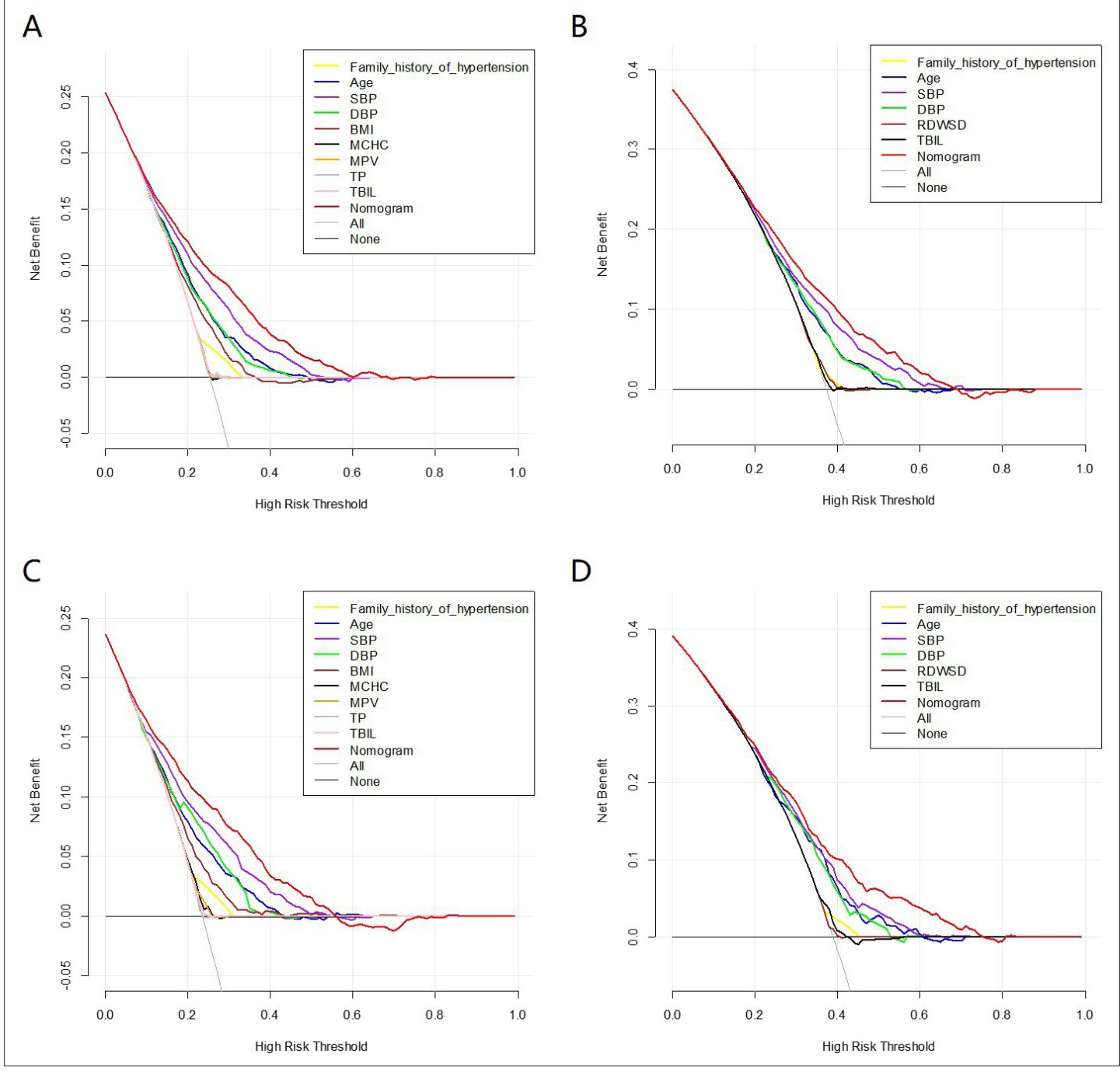

**Figure 5.** Decision curve analysis (DCA) of the nomogram prediction in the training set and validation set. (**A**) DCA of nomogram[140/90] prediction in the training set of Group[140/90]. (**B**) DCA of nomogram[130/80] prediction in the training set of Group[130/80]. (**C**) DCA of nomogram[140/90] prediction in the validation set of Group[140/90]. (**D**) DCA of nomogram[130/80] prediction in the validation set of Group[130/80]. SBP: systolic blood pressure; DBP: diastolic blood pressure; BMI: body mass index; MCHC: mean cell hemoglobin concentration; MPV: mean platelet volume; TP: total protein; TBIL: total bilirubin; TG: triglycerides; RDWSD: red blood cell distribution width standard deviation.

the risk of hypertension has been well documented in epidemiological and clinical studies (**Anderson et al., 2010**).

There are also some limitations in our study. Firstly, as mentioned above, the nomogram was constructed based on multivariate analysis of physical examination data between 2009 and 2019. Loss to follow-up and missing data reduced the effective sample size and may threaten the internal validity of the study. Secondly, some subjects who were diagnosed with secondary hypertension may

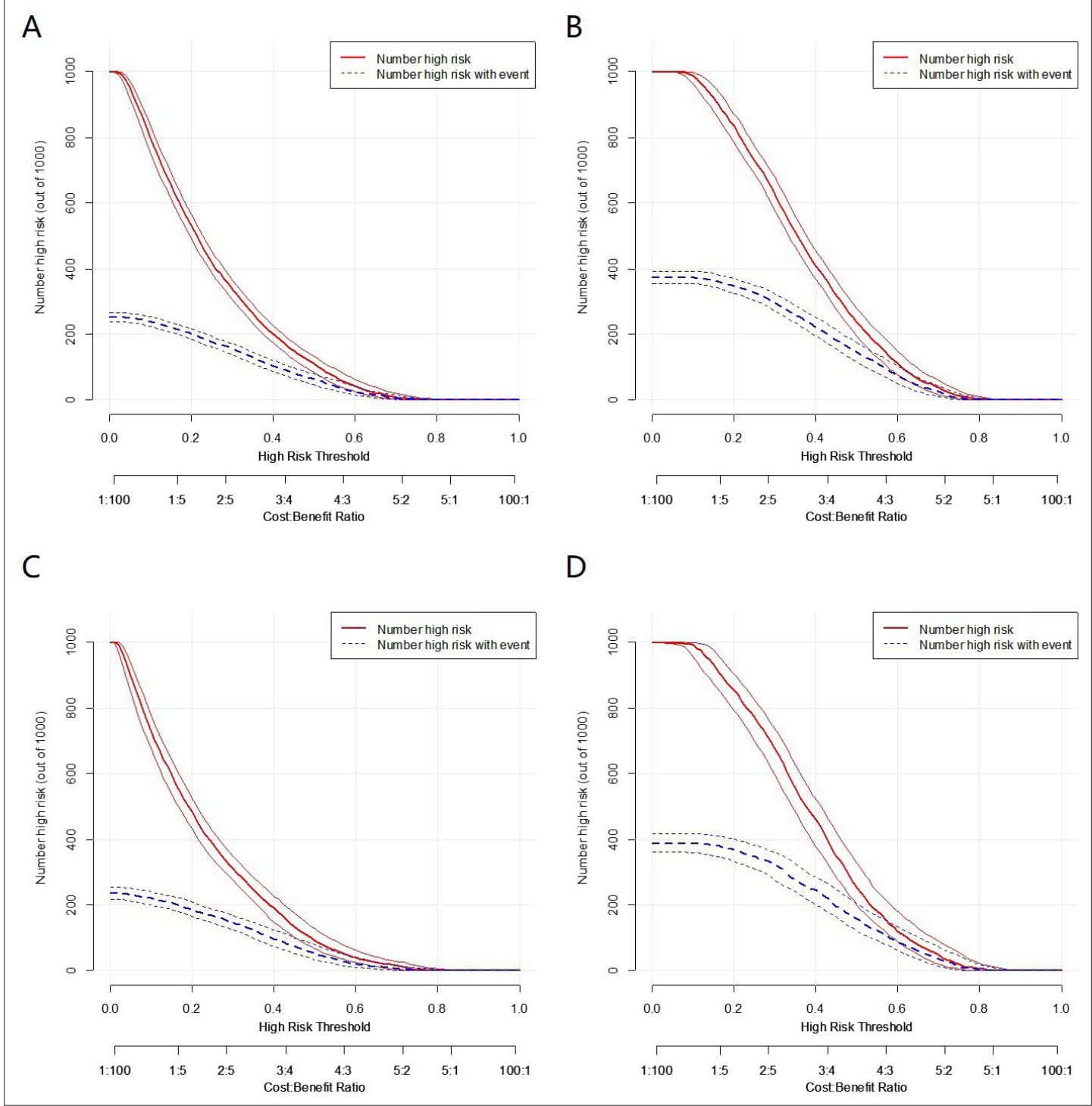

**Figure 6.** Clinical impact curves of the nomogram prediction in the training set and validation set. (**A**) Clinical impact curves of nomogram[140/90] prediction in the training set of Group[140/90]. (**B**) Clinical impact curves of nomogram[130/80] prediction in the training set of Group[130/80]. (**C**) Clinical impact curves of nomogram[140/90] prediction in the validation set of Group[140/90]. (**D**) Clinical impact curves of nomogram[130/80] prediction in the validation set of Group[130/80].

be also included in this study. The diagnosis of hypertension may lack strictness. Thirdly, the nomogram showed medium prediction accuracy may suggest that other factors should be included. Many blood parameters were deleted because of missing data. These may have inevitably caused bias. The prediction accuracy could perhaps be improved in further studies with large sample sizes and more

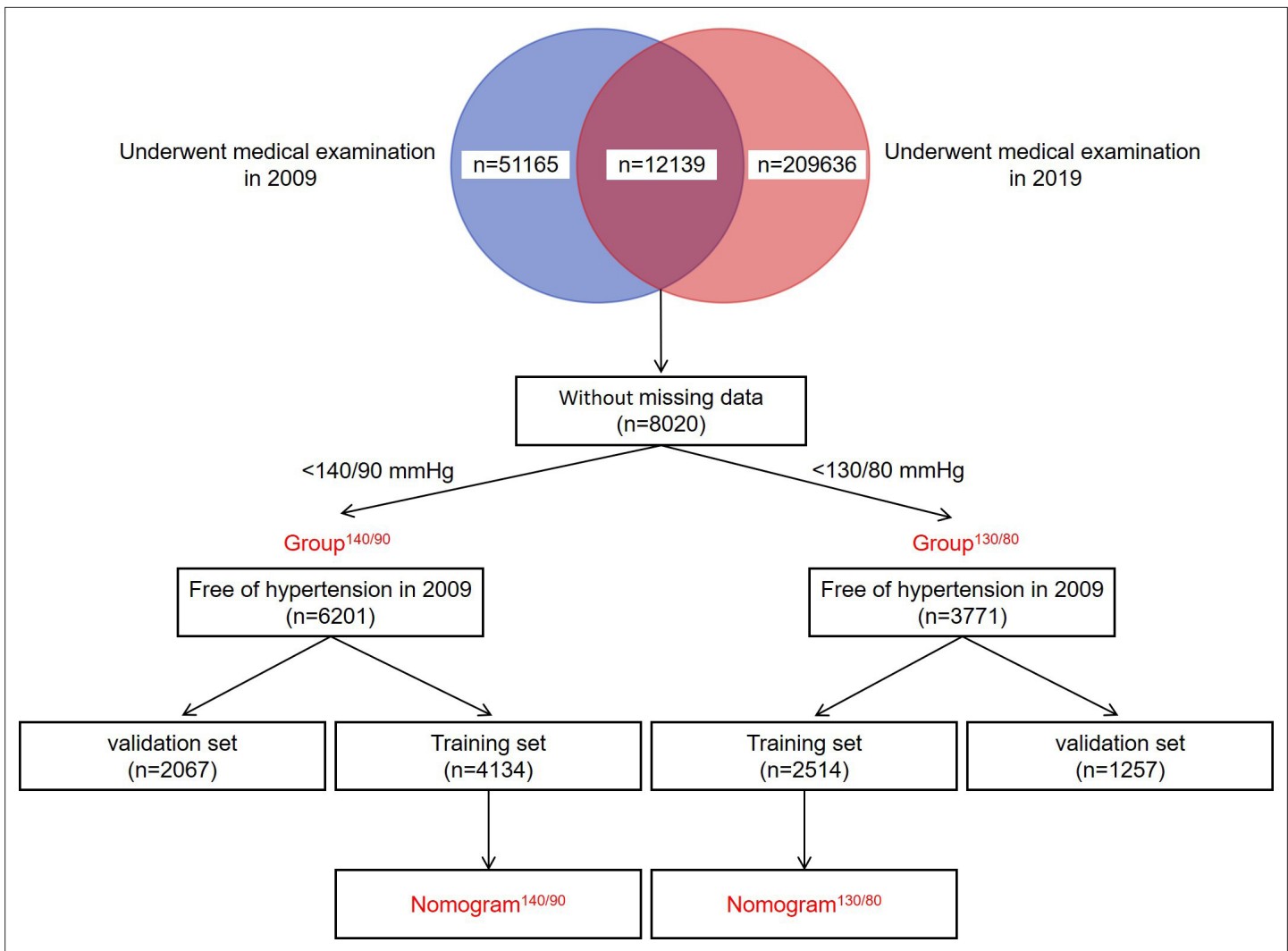

**Figure 7.** Flowchart of the procedure. A total of 51,165 and 209,636 subjects who underwent physical examination in 2009 and 2019 were enrolled in this study, respectively. 8020 subjects who underwent medical examination both in 2009 and 2019 were finally enrolled. At a cut-off value of 140/90 mmHg, 6201 subjects who had normal blood pressure in 2009 were enrolled in Group[140/90]. At a cut-off value of 130/80 mmHg, 3771 subjects who had normal blood pressure in 2009 were enrolled in Group[130/80]. The data of Group[140/90] and Group[130/80] were used to construct the nomogram[140/90] and nomogram[130/80] for predicting hypertension, respectively.

variables. Further multicenter external validation should be performed to verify the discriminating ability and generalizability of our nomogram.

## Materials and methods
### Study population and data collection

The current study was carried out based on a large cohort study, named the Physical Examination Survey. The survey was conducted among subjects who underwent medical examination in a physical examination center of Hebei Province in 2009 and 2019. As shown in *Figure 7*, a total of 51,165 and 209,636 subjects who underwent physical examination in 2009 and 2019 were enrolled in this study, respectively. To avoid potential observational bias, we excluded subjects who had taken antihypertensive drugs before the medical examination. Subjects who did not finish the procedures of this survey and had missing data on the collected parameters were also excluded. After rigorous screening, 8020 subjects who underwent medical examination both in 2009 and 2019 were finally enrolled. At a cut-off value of 140/90 mmHg, 6201 subjects who had normal blood pressure in 2009 were enrolled

in Group[140/90]. At a cut-off value of 130/80 mmHg, 3771 subjects who had normal blood pressure in 2009 were enrolled in Group[130/80]. The data of Group[140/90] and Group[130/80] were used to construct the nomogram[140/90] and nomogram[130/80] for predicting hypertension, respectively.

The socio-demographic and clinical parameters from the electronic medical records system were collected, including hypertension status in 2009, hypertension status in 2019, gender, family history of hypertension, smoking status, drinking status, age, SBP, DBP, height, weight, BMI, white blood cell count (WBC), lymphocyte count (LYMPH), neutrophil count (NEUT), lymphocyte percentage (LYMPHP), neutrophil percentage (NEUTP), red blood cell count (RBC), hemoglobin (HGB), hematocrit (HCT), mean corpuscular volume (MCV), mean corpuscular hemoglobin (MCH), MCHC, red blood cell distribution width-coefficient of variation (RDWCV), RDWSD, platelet count (PLT), MPV, plateletcrit (PCT), platelet distribution width (PDW), middle cell count (MID), middle cell percentage (MIDP), alanine aminotransferase (ALT), aspartate transaminase (AST), TP, albumin (ALB), total bilirubin (TBIL), direct bilirubin (DBIL), glucose (GLU), cholesterol (CHOL), TG, neutrophil-to-lymphocyte ratio (NLR), and platelet-to-lymphocyte ratio (PLR).

All procedures were approved by the Ethics Committee of Hebei General Hospital. All subjects' data were anonymized and de-identified prior to the analyses. The requirement for informed consent was therefore waived.

## Definition and assessment

Hypertension was defined as two diagnostic criteria: (1) SBP $\geq$ 140 mmHg or DBP $\geq$ 90 mmHg or antihypertensive medication use according to 2018 Chinese Guidelines and 2018 ESC/ESH guidelines; (2) SBP $\geq$ 130 mmHg or DBP $\geq$ 80 mmHg or antihypertensive medication use according to 2017 ACC-AHA guidelines. Blood pressure was measured after a minimum of 5 min rest in sitting position. BMI was computed by the ratio of body weight (kg) to height squared ($m^2$). The blood samples were collected in the morning on an empty stomach.

## Statistical Aanalysis

For construction and validation of the nomogram, the subjects were randomly divided into training set and validation set at a ratio of 2:1, respectively. The comparability between the two sets was then evaluated. Continuous variables with normal distribution were described as means ± standard deviation and analyzed with Student's $t$-test to infer the differences between the two sets. Continuous variables with skewed distribution were described as median (25% percentile, 75% percentile) and analyzed with Mann–Whitney U test. Categorical data were presented as numbers (percent) and analyzed with chi-square test or the Fisher's exact test for their comparisons.

The LASSO regression technique was used to select the optimal predictive features in the training set. Then, multivariate logistic regression analysis was used to identify the independent factors by incorporating the feature selected in the LASSO regression. Following the multivariate analysis, factors with a two-sided p value <0.05 were selected for developing the nomograms. The predictive accuracy of nomograms was measured by AUC of the ROC curve and concordance index (C-index) in both the training and validation sets. The consistency between the actual outcomes and predicted probabilities was measured by the calibration curve. The clinical utility of the nomograms was measured by DCA and clinical impact curves for a population size as 1000. To compare the predictive accuracy of the nomogram[140/90] with that of nomogram[130/80], NRI and IDI were calculated.

Statistical analyses were performed with R software version 4.0.0 (RRID:SCR_001905), SPSS version 24.0 (RRID:SCR_019096), and MedCalc 19.0.7 (RRID:SCR_015044). Two-sided p value < 0.05 was considered to be statistically significant.

## Conclusion

In conclusion, based on a 10-year retrospective cohort study, we developed and validated a simple and reliable nomogram to predict the risk of hypertension for the population of China. The nomogram demonstrated favorable predictive accuracy, discrimination, and clinical utility in the training set and validation set, indicating good performance in practical application. This visualization model and website will aid the patients and physicians to predict the 10-year risk of hypertension and better clinical management.

## Acknowledgements

This study was supported by the Hebei Science and Technology Department Program (no. H2018206110).

## Additional information

### Funding

| Funder | Grant reference number | Author |
|---|---|---|
| Hebei Science and Technology Department Program | H2018206110 | Haijiang Wu |

The funders had no role in study design, data collection and interpretation, or the decision to submit the work for publication.

### Author contributions

Xinna Deng, Conceptualization, Data curation, Formal analysis, Funding acquisition, Investigation, Methodology, Project administration; Huiqing Hou, Conceptualization, Data curation, Funding acquisition, Investigation, Methodology, Supervision, Validation, Visualization; Xiaoxi Wang, Conceptualization, Funding acquisition, Investigation, Methodology, Software, Supervision, Validation, Visualization; Qingxia Li, Conceptualization, Investigation, Software, Supervision, Validation, Writing – original draft, Writing – review and editing; Xiuyuan Li, Conceptualization, Project administration, Resources, Software, Supervision, Validation; Zhaohua Yang, Resources, Software, Supervision, Validation, Visualization, Writing – original draft, Writing – review and editing; Haijiang Wu, Conceptualization, Data curation, Formal analysis, Funding acquisition, Investigation, Methodology, Project administration, Resources, Software, Supervision, Validation, Visualization, Writing – original draft, Writing – review and editing

### Author ORCIDs

Haijiang Wu  http://orcid.org/0000-0003-1132-0352

### Ethics

After reviewing, this paper conforms to the principles of medical ethics. It is accepted to publish by Hebei General Hospital Ethics Committee (202043).

### Decision letter and Author response

Decision letter https://doi.org/10.7554/eLife.66419.sa1
Author response https://doi.org/10.7554/eLife.66419.sa2

## Additional files

### Supplementary files

• Transparent reporting form

### Data availability

Source data files have been provided.

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
