## [Decision Letter]

**Acceptance summary:**

In this work, the authors developed easy-to-use simple nomograms to predict the 10-year probability of hypertension from easily accessible health metrics. The study deposits an important dataset, and carries out classical multivariate analyses to arrive to a usable model. Not accounting for inter-dependencies may, however, limit the performance of the generated models. Going beyond nomograms and employing advanced, yet easily accessible, machine learning approaches may show the real potential of the compiled data.

**Decision letter after peer review:**

Thank you for submitting your article "Development and validation of a nomogram to better predict hypertension based on a 10-year retrospective cohort study" for consideration by *eLife*. Your article has been reviewed by 2 peer reviewers, and the evaluation has been overseen by a Reviewing Editor and Matthias Barton as the Senior Editor. The following individual involved in review of your submission has agreed to reveal their identity: Richard Woodman (Reviewer #1).

The reviewers and Editors have discussed their reviews with one another, and this letter will help you prepare a revised submission.

*Essential revisions:*

– The title would benefit from the mention of the population involved in the study, as the outcomes will likely not be transferrable to other populations. For the sake of clarity, it will also help if the manuscript body explicitly mentions that the predicted hypertension probabilities are for the 10-year risk.

– The search for independent features zooms in to 10 features for nomogram140/90 and 6 features for nomogram130/80. Biologically, any feature that has an association in defining the hypertension probability, should have a predictive relevance regardless whether the cutoff for the hypertension definition is 140/90 or 130/80. The used approach of feature trimming, and different numbers of features for the two nomograms, may neglect important interactions. Furthermore, the overall low number of features warrant the application of higher-end machine learning techniques for feature selection (if at all deemed to be necessary to eliminate any). For this, decision-tree-based techniques, such as random forests or gradient boosting machines, may have greater promise for arriving to better models, while internally producing feature importance rankings for possible elimination of the low performing ones (all done while accounting possible complex and multiplexed interactions between the features).

– The authors should explain their rationale for using the LASSO e.g. that they used it as a variable selection technique to reduce the number of parameters and thereby simply the risk prediction model? What was the degree of correlation amongst the 40 available covariates? Details of the package used in R to perform the LASSO should be provided. What was the approach used for the chosen value of lambda to identify the final LASSO model e.g. lambda that provides the minimum cross-validated MSE or lambda with minimum MSE +1SE? Were interactions and higher-order variables assessed in the LASSO. If not, why not?

– The calibration curve for BP 140/90 is, as the authors point out, poor especially for higher risk patients. This is a real concern if the model is to be used for accurate risk prediction in those at higher risk.

– It seems the NRI and IDI were used to determine the best model using an estimated risk prediction score from 130/80 versus 140/90 as the outcome. Normally the NRI and IDI are used to determine the value of additional covariates for risk prediction whereas here the authors are basing the choice of the outcome on the NRI and IDI. The authors should explain the logic of this and justify the approach to the Journal readers.

– How are the cases handled when only the SBP or the DBP falls beyond the cutoff, but not the pair? If excluded, can excluding such cases eliminate an important, high-risk subpopulation from the model development?

– The discriminatory power of the model is relatively modest. The authors should describe how the prediction accuracy could perhaps be improved.

– P-values to 5 decimal places are not necessary – consult the Journal guidelines.

– TG and TBIL, as predictors of hypertension, may be result of chance and the chosen sample and random sample. Except for the internal validation, the authors should discuss the need for external validation of the risk prediction model.

– Section 2.2, the second to last line. Written "weight (Kg)", should be "weight (kg)".

– Section 2.3, the C-index metric is introduced first time in the text (not counting the Abstract), without defining and expanding it.

– Section 3.3, written "… was assessed with the AUC and c-index", c should be capitalised for consistency.

– Section 4 (Discussion), 3rd paragraph. Written "… AUC of nomogram140/90 was higher than that of monogram140/90". The last one should be "nomogram130/80".

– In the same section, written "Similar to our study, the Iranian research from revealed the same result…". Seems there is a missing word in between "from" and "revealed".

– Figure 1 caption. Needs an expansion with a brief description of the content.

– Figure 2. The x-axes in A and C are labelled as Log(λ), while those for B and D are labelled as Log Lambda. Please, change them to be the same.

– Figure 2 caption. The caption needs comments about the line colours and about the lines that stand out in B and D.

– Figure 3 and the associate text will benefit from a brief description of how one should use the nomogram. It is easy to infer from the nomogram itself, but, considering the presence of multiple types of nomograms, explicitly describing the usage of this particular type will save a few minutes for the readers.

– Figure 4. Please organise the line labelling brought next to the plots in an ascending or descending order for the corresponding AUC values.

– Most tables are missing footnotes to describe the abbreviations.

---

## [Author Response]

Essential revisions:– The title would benefit from the mention of the population involved in the study, as the outcomes will likely not be transferrable to other populations. For the sake of clarity, it will also help if the manuscript body explicitly mentions that the predicted hypertension probabilities are for the 10-year risk.

Thank you for your suggestion. We have changed the title of the article to “Development and validation of a nomogram to better predict hypertension based on a 10-year retrospective cohort study in China”. In addition, for the sake of clarity, the sections of abstract, discussion and conclusion of the article have mentioned that the study was carried out based on a 10-year retrospective cohort and the predicted hypertension probabilities are for the 10-year risk.

– The search for independent features zooms in to 10 features for nomogram140/90 and 6 features for nomogram130/80. Biologically, any feature that has an association in defining the hypertension probability, should have a predictive relevance regardless whether the cutoff for the hypertension definition is 140/90 or 130/80. The used approach of feature trimming, and different numbers of features for the two nomograms, may neglect important interactions. Furthermore, the overall low number of features warrant the application of higher-end machine learning techniques for feature selection (if at all deemed to be necessary to eliminate any). For this, decision-tree-based techniques, such as random forests or gradient boosting machines, may have greater promise for arriving to better models, while internally producing feature importance rankings for possible elimination of the low performing ones (all done while accounting possible complex and multiplexed interactions between the features).

Thank you for raising these important issues. (1) Indeed, the human body is a complex biological network. Holistically, all biological features may contribute to or be influenced by blood pressure as well as hypertension. The current study aimed to identify statistically significant variables from aspect of socio-demographic and clinical characteristics associated with hypertension, and subsequently developed a simple, reliable nomogram to better predict hypertension probabilities. As mentioned in the article, the nomogram showing medium prediction accuracy may suggest that other factors should be included. Further extensive studies to focus on this issue are needed in the future. (2) We totally agree with your suggestion that decision-tree-based techniques (such as random forests or gradient boosting machines) may have greater promise for finding more useful information contained in variables. The current study aimed to develop a simple nomogram to predict hypertension probabilities. By reducing the number of variables in a model, we can reduce overfitting and the complexity of the model, make it easier to interpret, and decrease operation complexity. Lasso is useful because it is a shrinkage estimator: it shrinks the size of the coefficients of the independent variables depending on their predictive power. Some coefficients may shrink down to zero, allowing us to restrict the model to variables with nonzero coefficients. We performed a systematic review of published literature on PubMed. With the search strategy “(nomogram[Title/Abstract]) AND (lasso[Title/Abstract])”, 594 articles were found. With the search strategy “(nomogram[Title/Abstract]) AND (lasso[Title/Abstract])”, 53 articles were found. However, with the search strategy “(nomogram[Title/Abstract]) AND (gradient boosting machine[Title/Abstract])”, Only 1 article was found. These results may imply that LASSO Cox regression is a widely-used method for high-dimensional predictors selection and nomogram construction. Of course, if you feel that decision-tree-based techniques are essential for this article, we would be willing to carry out the additional experiments.

– The authors should explain their rationale for using the LASSO e.g. that they used it as a variable selection technique to reduce the number of parameters and thereby simply the risk prediction model? What was the degree of correlation amongst the 40 available covariates? Details of the package used in R to perform the LASSO should be provided. What was the approach used for the chosen value of lambda to identify the final LASSO model e.g. lambda that provides the minimum cross-validated MSE or lambda with minimum MSE +1SE? Were interactions and higher-order variables assessed in the LASSO. If not, why not?

Thank you for raising these important issues.

(1) As mentioned above, the current study aimed to develop a simple nomogram to predict hypertension probabilities. By reducing the number of variables in a model, we can reduce overfitting and the complexity of the model, make it easier to interpret, and decrease operation complexity. Lasso is useful because it is a shrinkage estimator: it shrinks the size of the coefficients of the independent variables depending on their predictive power. Some coefficients may shrink down to zero, allowing us to restrict the model to variables with nonzero coefficients (factors that affecting outcomes).

(2) The degree of correlation amongst the 40 available covariates are listed in Author response image 1.

**Author response image 1. sa2fig1:** 

(3) Details of the package used in R to perform the LASSO are listed below:library(glmnet)

library(plotmo)

setwd("C:\\Users\\1")

mydata<-data.frame(read.csv(file = "input.txt",sep = "\t",header = TRUE))

head(mydata)

dim(mydata)

summary(mydata)

v1<-as.matrix(mydata[,c(3:42)])

v2 <-mydata[,2]

myfit <- glmnet(v1, v2, family = "binomial")

lam=myfit$lambda

summary(log2(lam))

pdf(file="lambda.pdf",width=6,height=6)

plot_glmnet(myfit, xvar = "lambda", label = TRUE)

dev.off()

myfit2$lambda.min

coe <- coef(myfit, s = myfit2$lambda.min)

act_index <- which(coe ! = 0)

act_coe <- coe[act_index]

row.names(coe)[act_index]

coe

(4) MSE was separately calculated for each variable. Lasso provides a minimum MSE of 0.003713099 on the data. The minimum MSE was used as cutoff to evaluate predictive performance. (5) The interactions and higher-order variables were not assessed in the LASSO. In the article, the aim of LASSO regression technique is to select the optimal predictive features by using the minimum MSE criteria.

– The calibration curve for BP 140/90 is, as the authors point out, poor especially for higher risk patients. This is a real concern if the model is to be used for accurate risk prediction in those at higher risk.

Thank you for raising this important issue. As shown in Figure 4, both in the training set and validation set, the calibration curves showed a good agreement between nomogram-predicted probability and the actual outcome. Although the nomogram-derived curve may overestimate the probability by about 10% for higher risk patients, this model still had a good calibration in the internal validation cohort. As discussed in the article, this bias may be negligible in further studies with large sample sizes and more variables.

– It seems the NRI and IDI were used to determine the best model using an estimated risk prediction score from 130/80 versus 140/90 as the outcome. Normally the NRI and IDI are used to determine the value of additional covariates for risk prediction whereas here the authors are basing the choice of the outcome on the NRI and IDI. The authors should explain the logic of this and justify the approach to the Journal readers.

Thank you so much for your professional suggestion. Indeed, the NRI and IDI were originally proposed to characterize accuracy improvement in predicting a binary outcome, when new biomarkers are added to regression models. Most recently, these two indices have been extended from binary outcomes to multi-categorical and survival outcomes (PMID of reference: 33324980). For example, in 2020, Zhang et al. compared the predictive ability of a stroke prediction model (China-PAR) with the revised Framingham Stroke Risk Score (R-FSRS) for 5-year stroke incidence in a community cohort of Chinese adults (PMID of reference: 33192957). The two prediction models have 5 same risk factors, as well as 2 additional factors in R-FSRS and 6 additional factors in China-PAR. The NRI and IDI values were assessed to compare the discrimination ability of two prediction models. Similarly, to better assess the performance of nomogram^140/90^ and nomogram^130/80^, the NRI and IDI were also used to determine the best model. According to your suggestion, the above discussion have been added in the article.

– How are the cases handled when only the SBP or the DBP falls beyond the cutoff, but not the pair? If excluded, can excluding such cases eliminate an important, high-risk subpopulation from the model development?

Thank you so much for your professional suggestion. According to the 2017 American College of Cardiology/American Heart Association guideline, arterial hypertension is defined as systolic blood pressure (SBP) equal to or greater than 140 mm Hg or diastolic blood pressure (DBP) equal to or greater than 90 mm Hg. Similarly, the cases whose SBP or DBP fell beyond the cutoff in 2009 can be explicitly diagnosed as hypertension. The current study aimed to develop a nomogram based on the cases who had normal blood pressure in 2009. Therefore, we specifically excluded the cases who had abnormal blood pressure.

– The discriminatory power of the model is relatively modest. The authors should describe how the prediction accuracy could perhaps be improved.

Thank you so much for your professional suggestion. As mentioned in in the Discussion section of our revised manuscript, there are also some limitations in our study. Firstly, the nomogram was constructed based on multivariate analysis of physical examination data between 2009 and 2019. Loss to follow-up and missing data reduced the effective sample size and may threaten the internal validity of the study. Secondly, some subjects who were diagnosed with secondary hypertension may be also included in this study. The diagnosis of hypertension may lack strictness. Thirdly, the nomogram showing medium prediction accuracy may suggest that other factors should be included. Many blood parameters were deleted because of missing data. These may have inevitably caused bias. The prediction accuracy could perhaps be improved in further studies with large sample sizes and more variables. According to your suggestion, the above discussion have been added in the article.

– P-values to 5 decimal places are not necessary – consult the Journal guidelines.

Thanks for your careful checks. According to your suggestion, we have corrected the P-values to 4 decimal places.

– TG and TBIL, as predictors of hypertension, may be result of chance and the chosen sample and random sample. Except for the internal validation, the authors should discuss the need for external validation of the risk prediction model.

Thank you so much for your professional suggestion. The current study aimed to identify statistically significant variables from aspect of socio-demographic and clinical characteristics associated with hypertension, and subsequently developed a simple, reliable nomogram to better predict hypertension probabilities. According to your suggestion, we have discussed the need for external validation of the risk prediction model. In addition, we would like to carry out a separate and extensive study to focus on this issue in the near future.

– Section 2.2, the second to last line. Written "weight (Kg)", should be "weight (kg)".

According to your suggestion, we have corrected the “weight (Kg)” into “weight (kg)".

– Section 2.3, the C-index metric is introduced first time in the text (not counting the Abstract), without defining and expanding it.

According to your suggestion, we have corrected the “C-index” into “concordance index (C-index)”.

– Section 3.3, written "… was assessed with the AUC and c-index", c should be capitalised for consistency.

According to your suggestion, we have corrected the “c-index” into “C-index”.

– Section 4 (Discussion), 3rd paragraph. Written "… AUC of nomogram140/90 was higher than that of monogram140/90". The last one should be "nomogram130/80".

According to your suggestion, we have corrected the “monogram^140/90^” into “monogram^130/80^”.

– In the same section, written "Similar to our study, the Iranian research from revealed the same result…". Seems there is a missing word in between "from" and "revealed".

According to your suggestion, we have reformulated the previous sentence as followed: “Similar to our study, researchers from Iran revealed the same result, reporting that sex was not found to be an independent risk factor for hypertension.”

– Figure 1 caption. Needs an expansion with a brief description of the content.

Thank you for your suggestion, and we have made changes accordingly. A brief description of Figure 1 has been added followed the caption.

– Figure 2. The x-axes in A and C are labelled as Log(λ), while those for B and D are labelled as Log Lambda. Please, change them to be the same.

According to your suggestion, we have corrected the “Log(λ)” into “Log Lambda”.

– Figure 2 caption. The caption needs comments about the line colours and about the lines that stand out in B and D.

According to your suggestion, we have added the legends to Figure 2B and Figure 2D. In addition, we have added comments about the lines that stand out in B and D in the figure caption.

– Figure 3 and the associate text will benefit from a brief description of how one should use the nomogram. It is easy to infer from the nomogram itself, but, considering the presence of multiple types of nomograms, explicitly describing the usage of this particular type will save a few minutes for the readers.

According to your suggestion, we have added a brief description of how one should use the nomogram in the figure caption.

– Figure 4. Please organise the line labelling brought next to the plots in an ascending or descending order for the corresponding AUC values.

According to your suggestion, we have organized the line labelling brought next to the plots in a descending order for the corresponding AUC values.

– Most tables are missing footnotes to describe the abbreviations.

According to your suggestion, the abbreviations have been added in the footnotes of all tables.